# Drivers of malnutrition among late adolescent and young women in rural Pakistan: a cross-sectional assessment of the MaPPS trial

Jo-Anna B Baxter [ID],[1,2] Yaqub Wasan,[3] Amjad Hussain,[3] Sajid B Soofi [ID],[3] Imran Ahmed,[3] Zulfiqar A Bhutta[1,2,3]

[1]Centre for Global Child Health, The Hospital for Sick Children, Toronto, Ontario, Canada
[2]Department of Nutritional Sciences, University of Toronto, Toronto, Ontario, Canada
[3]Centre of Excellence in Women and Child Health, Aga Khan University, Karachi, Pakistan

**Correspondence to**
Professor Zulfiqar A Bhutta;
zulfiqar.bhutta@aku.edu

## ABSTRACT

**Objective** This study aimed to characterise the burden of malnutrition and assess how underlying determinants at the structural and intermediary levels contributed to malnutrition among late adolescent and young women in rural Pakistan.

**Design** Cross-sectional enrolment data assessment.

**Setting and participants** This study was conducted using data from adolescent and young women (n=25 447) enrolled in the Matiari emPowerment and Preconception Supplementation Trial, collected from June 2017 to July 2018 in Matiari District, Pakistan. The WHO-based cut-offs were applied to anthropometric measures to estimate body mass index (BMI) categories (underweight, overweight, obese) and stunting. Hierarchical models were generated to evaluate the association between the determinants with BMI categories and stunting among late adolescent girls and young women, respectively.

**Primary and secondary outcome measures** The main outcomes of interest were BMI categories and stunting. Explanatory variables included measures of socioeconomic status, education, occupation, health, well-being, food security, empowerment and food practices.

**Results** Regardless of age group, the prevalence of underweight was high (36.9%; 95% CI 36.3% to 37.5%). More late adolescent girls were underweight, while more young women were overweight/obese (p<0.001). Stunting affected 9.2% (95% CI 8.9% to 9.6%) of participants, of which 35.7% were additionally underweight and 7.3% overweight/obese. Compared with those in the normal weight category, those underweight were more likely to be impoverished and less empowered. Those overweight/obese were more likely to be from a higher wealth quintile and food secure. Increased education level and food security were associated with reductions stunting risk.

**Conclusions** This study informs the data gap and need for comprehensive research on adolescent nutritional status. Findings suggest factors related to poverty played an important, underlying role in undernutrition among participants. Commitment to improving the nutritional status of all adolescent and young women in Pakistan will be critical given the observed burden of malnutrition.

**Trial registration number** NCT03287882.

### STRENGTHS AND LIMITATIONS OF THIS STUDY

⇒ Although the catchment area of this study was restricted to the areas covered by the Lady Health Worker programme, the study is very large and representative of rural setting in Pakistan.

⇒ Given the cross-sectional design of the study, we cannot suggest causation.

⇒ All anthropometric measurements (weight, height, middle upper arm circumference) were collected in duplicate following standardised procedures by trained study personnel, although we did not collect additional measures of body composition (eg, skinfold thickness, abdominal circumference).

⇒ We used validated questionnaires to assess social determinants of health at both the structural and intermediary levels.

### INTRODUCTION

Adolescence is a life stage characterised by substantial physical, cognitive, social and emotional development, and nutritional status importantly underlies the transition from childhood to adulthood.[1] Defined by the WHO as the period from 10 to 19 years, some suggest adolescence extends to 24 years given continued cortical development.[1 2] Being malnourished during the adolescent period is associated with poor health outcomes throughout the life course and has potential intergenerational effects.[3 4] In South Asia, undernutrition affects >38% of adolescent girls; there has been limited decline in the past decade and the increasing occurrence of overweight/obesity is of additional concern.[5]

Anthropometric measures can serve as a useful indicator for assessing malnutrition. Measures of undernutrition, including underweight (thinness) and stunting (short stature), are determined from body mass index (BMI) and attained height, respectively. While underweight reflects current nutritional status caused by insufficient or

imbalanced intake and/or underlying disease, stunting reflects chronic or recurrent undernourishment.[6] Being underweight and/or stunted has been found to be associated with adverse health consequences to the individual, and, in the event of a pregnancy, increased risk for adverse pregnancy outcomes.[7] Conversely, overweight and obesity have been linked to the early onset of non-communicable diseases and premature mortality.[8 9]

Social determinants of health (SDoH) are described to influence nutritional inequalities, making understanding their role in malnutrition of importance.[10] SDoH are typically divided into two classes, structural and intermediary.[11] Structural SDoH underlie food availability and access, including stratifiers such as ethnicity, education, occupation and income. Intermediary SDoH are more proximal to nutrition-related outcomes, affecting material circumstances and biological, psychological and behavioural factors.[12]

With the focus of the Sustainable Development Goals on equity, and the explicit aim in goal 5 to empower all women and girls,[13] measuring, monitoring and researching women's empowerment is of importance. Adolescent girls are an understudied population with respect to empowerment, and understanding the impact of interventions that aim to improve empowerment is a noted research gap.[1] Proxies for empowerment span SDoH at both the structural (eg, education) and intermediary levels (eg, self-efficacy and contribution to decision-making). They are suggested to influence nutritional status, although this association has predominantly been made between maternal empowerment and child nutritional status.[3 14 15] The WHO acknowledges that empowering adolescent girls, families and communities is an important component to improving overall health and nutritional status.[1]

At this time, there are more young people than ever before in Pakistan, with around 30% of the population between 15 and 24 years.[16] Nationally representative surveys have traditionally focused on married women 15–49 years given the importance of reproductive outcomes, and estimate that the prevalence of underweight and stunting are high at 8.5% and 5.0%, respectively, and that there is an emerging burden of overweight and obesity.[16] However, there has been limited assessment among late adolescent girls and unmarried young women, thus there is a gap in information within this segment of the population. We aimed to assess the prevalence of malnutrition using anthropometric measures from all adolescent and young women (15–23 years) living in rural Pakistan, and determine whether structural and intermediary SDoH were associated with underweight, overweight/obese and stunting. Understanding the diverse SDoH that predict malnutrition will be informative to policy generation and the prioritisation of programming to address current nutritional challenges.

## METHODS
This study used cross-sectional data from late adolescent girls and young women (15–23 years) collected at enrolment within the Matiari emPowerment and Preconception Supplementation (MaPPS) trial.

### Setting and study design
The MaPPS trial is a two-arm, cluster-randomised, controlled trial that aims to assess the impact of life skills building education and multiple micronutrient supplementation on several nutrition, health and empowerment-related outcomes in rural Pakistan. The study methodology and data collection tools employed within this population-based effectiveness trial have been previously described.[17] The trial was registered at ClinicalTrials.gov as NCT03287882. The MaPPS trial aimed to recruit approximately 25 400 non-pregnant adolescent and young women to observe enough pregnancies to detect a 25% relative reduction in low birthweight births (primary trial outcome), assuming prevalence of low birth weight of 30%, intra-class correlation coefficient (ICC)=0.011, k=0.16, accounting for 10% attrition, and probability of type I and type II errors of 0.05 and 0.20, respectively.

### Participant enrolment
Participants were recruited to the MaPPS trial from the end of June 2017 to July 2018, using a clusterwise roll-out. The cluster unit was the catchment population of preidentified health facilities and their respective outreach community health workers (Lady Health Workers; LHWs) (n=26). Late adolescent girls and young women 15–23 years were identified from a household listing exercise conducted from December 2016 to May 2017, wherein all households covered by LHWs in a cluster unit were surveyed by study personnel. To screen for participation in the MaPPS trial, late adolescent girls and young women 15–23 years were approached at their homes by study personnel to further assess their eligibility and ≥1 participant could be enrolled per household. Eligibility criteria included reporting being non-pregnant; physically able to comply with supplementation; not participating in any other nutrition studies; and intending to remain in the study area for the duration of the trial. The purpose and voluntary nature of the trial was verbally explained following confirmation of eligibility by study personnel and a written consent form was provided. Written documentation of informed consent was obtained as either a signature or a thumb impression, in the case that a participant was illiterate.

### Patient and public involvement
Prior to the start of the household listing exercise, a wide range of activities were undertaken to obtain local approval of the MaPPS trial. This included sharing information about what was being done within the MaPPS trial and why with the district education department and health leaders (including the LHW programme), and leaders from participating villages. The broader community and prospective participants were not invited to comment

on the study design, and they were not consulted on the study outcomes or result interpretation for this research.

## Data collection

On enrolment, a questionnaire was administered to collect information on participant demographics; socioeconomic status; reproductive health; general health, well-being and nutrition; and empowerment-related variables. Anthropometric (height, weight, middle upper arm circumference) measurements were collected in duplicate by two trained data collectors using standardised methods.[18] Preset allowable differences were employed (height: <1.0 cm; weight: <0.5 kg), and a third measurement was collected if two measurements exceed this. In all analyses, the average of two acceptable measures was used. The technical error of measurement (TEM), relative TEM (%TEM) and coefficient of reliability (R) were calculated for height and weight.[19] Field monitoring teams additionally conducted regular data quality spot checks.[17] Data collectors were blinded to study arm allocation.

## Outcomes of interest

The outcomes of interest in this study were cut-offs for BMI (underweight: <18.5 kg/m$^2$, normal: 18.5–24.9 kg/m$^2$, overweight: 25–29.9 kg/m$^2$, obese: ≥30 kg/m$^2$) and stunting (height <145 cm).[20] BMI was considered categorically because of the increased health risks when it is too low or high. There are known issues with the adolescent-specific cut-offs for BMI and stunting (10–19 years),[21 22] thus we applied the WHO cut-offs for women of reproductive age (WRA; 15–49 years). This is common in international reporting given the age range overlap. Furthermore, the adult cut-offs are tied to reproductive outcomes and adolescent pregnancies were common in the study population.[16] For completeness, since the WHO Growth Reference is suggested to classify the growth of adolescents up to 19 years (228 completed months), we also applied the WHO igrowup package for Stata (StataCorp) to adolescents' anthropometric data to determine height-for-age and BMI-for-age z-scores (HAZ and BAZ, respectively).[23]

## Explanatory variables

Multiple explanatory variables were hypothesised to underlie the nutritional status outcomes of interest. The selection of underlying SDoH risk factors followed our previously published conceptual framework,[24] which was guided by the UNICEF framework on the causes of malnutrition[25] and the WHO framework on SDoH related to health and well-being.[26] These were split into two categories: structural (education, occupation, socioeconomic status and religion) and intermediary (health, well-being, food insecurity, empowerment and food practices).

Indicators for demographic information, socioeconomic status and reproductive health were adapted from the Pakistan Demographic Health Survey (PDHS).[16] A variable for wealth quintile was derived using a principal

component analysis from factors related to home characteristics and household asset ownership.[16] Other health-related proxies included self-reported health status and mental health (experience of depression, anxiety and stress-related emotions over the preceding week; Depression, Anxiety and Stress Score-21 items).[27] Other nutrition-related proxies included household food insecurity (Household Food Insecurity Access Scale)[28] and skipping breakfast and sharing dinner with family (Health Behaviour in School-aged Children Survey).[29] Empowerment proxies included generalised self-efficacy (Generalised Self-Efficacy Scale)[30] and decision-making autonomy (PDHS).[16] Self-efficacy scores were categorised into tertiles, corresponding to low, moderate and high. A variable for decision-making autonomy was derived from reported participation in five decisions, adapted from the PDHS (decisions: own food consumption, own healthcare, household food purchases, household food distribution, household purchases). The scoring of decision-making participation was consistent with the Survey-based Women's Empowerment Index (SWPER) for WRA[31] and participants were categorised based on their involvement (all decisions made by family, most decisions made by family, decisions made jointly with family or autonomously).

## Statistical analyses

Analyses were performed using Stata, V.15.0 (Stata). Since the enrolment data were collected prior to the study intervention, data were analysed cross-sectionally, independent of study arm assignment. To describe the study population, we used descriptive statistics, including means and SD and proportions for continuous and categorical variables, respectively. To be consistent with the standard age-based reporting of adolescent anthropometric measures, demographic and anthropometric outcomes were further disaggregated by age group (late adolescent girls: 15–18.9 years; young women: 19–23 years), and a $\chi^2$ test or t-test was used to determine whether there was a difference between proportions and continuous measures, respectively.

We aimed to generate a hierarchical, multivariable model to examine the associations between the SDoH explanatory variables and BMI categories (base: normal BMI (18.5–24.9 kg/m$^2$)) and stunting (base: height >145 cm), respectively, using our conceptual framework as a guide. The categories for overweight and obese were combined for the BMI variable, given the low prevalence of obesity observed. Because we found that there was a difference in the prevalence of both BMI categories and stunting between late adolescent girls and young women, and to be consistent with standard age-based reporting of anthropometric measures, a separate model was generated for each age group.

The multilevel model building approach described by Victora et al[32] was followed to generate an adapted hierarchical model with four levels. Guided by our conceptual model,[24] we hypothesised that there were four levels

of SDoH explanatory variables within the hierarchy and mapped corresponding variables: (1) socioeconomic status (distal-most level): education, occupation, religion and wealth quintile; (2) household and personal factors: marital status, parity and household food security; (3) health and well-being: perception of own health, experience of depression-like, anxiety-like and stress-like feelings and (4) actions and practices (proximal-most level): self-efficacy, participation in decision making, skipping breakfast and eating dinner with family. For all analyses, the reference for each SDoH explanatory variable was set as the category hypothesised to present the greatest nutritional vulnerability.[24] To investigate the association between the SDoH explanatory variables and each primary outcome measure, crude analyses were first conducted using logistic regression. Variables were considered potentially relevant for inclusion in the multivariable hierarchical model if the unadjusted p was <0.200.[33] Starting with the distal-most level, relevant variables were entered into the model. At each level, a backward-stepwise model building approach was used. Variables were serially removed to observe any change in the β-coefficients of those variables maintained in the model and find a model that best explained the variability in the data. When the p of an explanatory variable was <0.05, it was retained in the final model. If the p was ≥0.05 and/or the SE was high, exclusion of the variable was explored. If the β-coefficients of other variables in the model changed by >10%, the excluded variable was returned, otherwise the variable was dropped. Multicollinearity between variables was assessed using the variance inflation factor (VIF), although we did not find that multicollinearity affected the selected variables using the criteria VIF ≥5. The effect estimates in the final model for retained variables correspond to the point of addition to the model. Effect estimates are presented as relative risk ratios for the variable for BMI categories (three possible outcomes) and OR for stunting (two possible outcomes). Given the cluster-randomised design of the MaPPS trial, all models employed cluster robust SEs to account for possible correlation between clusters.

## RESULTS

### Enrolment and exclusion

Of the 32 141 adolescent and young women 15–23 years identified from the household listing exercise, 28 195 (87.7 %) were located, and 25 447 (79.2%) consented to participate in the trial and completed the enrolment questionnaire. Among those enrolled, 14 771 (58.0%) were 15–18.9 years and 10 676 (42.0%) were 19–23 years, representing 17 044 households.

### Participant demographic and reproductive characteristics

At the time of enrolment, participants' mean age (±SD) was 18.8±2.3 years (table 1). No formal education was reported by nearly half of participants, and those with schooling had completed 7.7±3.3 years on average.

Currently being a student was reported by 13.8% of participants, of which only 61.1% reported regular attendance. Employment outside the home was reported by 38.5% of participants. Common occupations were unskilled and skilled manual labour (17.5% and 20.2% of participants, respectively), although the majority worked within the home (47.5%). Nearly all participants had experienced menarche (25 305 (99.4%)), with the mean age of onset at 13.0±0.9 years (range: 10–21 years). 22.3% of participants were married, with the mean age at marriage of 16.9±1.8 years (range: 13–23 years) and 13.7% had been pregnant. On average, those married were 4.8±4.2 years younger than their husbands.

### Anthropometric characteristics

Regardless of age group, the majority of participants were of 'normal' body weight (52.9%; table 1), although the prevalence of underweight was high (36.9%). More late adolescent girls appeared to be underweight, while more young women were overweight/obese (p<0.001). Height was <145 cm among 9.2% of participants (late adolescent girls: 10.2%, young women: 7.9%; p<0.001) and 35.7% of those stunted were additionally underweight (late adolescent girls: 39.6%, young women: 28.7%; p<0.001) and 7.3% overweight/obese (late adolescent girls: 5.4%, young women: 10.7%; p=0.006). The TEM for weight and height were 0.22 and 0.18, respectively; %TEM were 0.47% and 0.12%; and the R for each measure was >99%, representing high reliability between the repeat measurements. Among adolescent participants <19 y, HAZ and BAZ were found to be −1.58±0.83 and −0.67±1.13, respectively.

### Hierarchical model generation

#### BMI categories and SDoH risk factors among late adolescent girls

For the hierarchical model looking at BMI categories among late adolescent girls, of the SDoH risk factors identified from the crude analyses, those maintained in the final model included occupation, wealth quintile, marital status, food insecurity, reported perception of health and self-efficacy (p for model: <0.0001; table 2). While a crude association was observed between BMI categories with education level, religion, decision-making and skipping breakfast, respectively, this was not maintained in the multivariable model (see online supplemental table S1 for all associations and distribution of responses for possible determinants). Compared with those with a normal BMI, those underweight were more likely to be from a poorer wealth quintile, unmarried, food insecure, report poorer overall health and have lower self-efficacy. Among those overweight/obese, compared with those with a normal BMI, someone was less likely to be employed as an unskilled manual labourer and unmarried, and more likely to be from a higher wealth quintile, food secure and report good health.

#### BMI categories and SDoH risk factors among young women

Among young women, SDoH risk factors maintained in the final hierarchical model for BMI included occupation,

**Table 1** Demographic and anthropometric characteristics among MaPPS Trial participants at enrolment, disaggregated by age group*

| Characteristic | All participants (n=25 447) | 15–18.9 years (n=14 771) | 19–23 years (n=10 676) | P value |
|---|---|---|---|---|
| Age (years) | 18.8±2.3 | 17.1±1.2 | 21.1±1.2 | <0.001 |
| Highest level of education completed | | | | <0.001 |
| None | 11 384 (44.7) | 4572 (42.0) | 2758 (44.5) | |
| Primary | 6063 (23.8) | 2721 (25.0) | 1412 (22.8) | |
| Secondary or higher | 8000 (31.5) | 4822 (32.6) | 3178 (29.8) | |
| Employed outside the home | 9620 (38.1) | 5395 (36.6) | 4225 (40.1) | <0.001 |
| Muslim | 23 045 (90.6) | 13 288 (90.0) | 9757 (91.4) | <0.001 |
| Wealth quintile | | | | <0.001 |
| Poorest | 4549 (17.9) | 2666 (18.0) | 1883 (17.6) | |
| Poor | 4853 (19.1) | 2881 (19.5) | 1972 (18.5) | |
| Middle | 5139 (20.2) | 3025 (20.5) | 2114 (19.8) | |
| Rich | 5360 (21.1) | 3145 (21.3) | 2215 (20.7) | |
| Richest | 5546 (21.8) | 3054 (20.7) | 2492 (23.3) | |
| Currently married† | 5685 (22.3) | 1418 (9.6) | 4267 (40.0) | <0.001 |
| Age when married husband (y)‡ | 16.9±1.8 | 15.9±1.1 | 17.2±1.8 | <0.001 |
| Has been pregnant§ | 3478 (13.7) | 572 (3.9) | 2906 (27.2) | <0.001 |
| Age at first pregnancy¶ | 17.5±1.6 | 16.4±1.1 | 17.7±1.6 | <0.001 |
| Weight (kg) | 47.0±9.4 | 45.6±8.5 | 48.9±10.2 | <0.001 |
| Height (cm) | 152.5±5.7 | 152.0±5.6 | 152.7±5.8 | <0.001 |
| BMI (kg/m$^2$) | 20.2±3.7 | 19.7±3.3 | 20.9±4.0 | <0.001 |
| BMI categorisation | | | | <0.001 |
| Underweight (<18.5 kg/m$^2$) | 9396 (36.9) | 6105 (41.3) | 3291 (30.8) | |
| Normal (18.5–24.9 kg/m$^2$) | 13 463 (52.9) | 7597 (51.4) | 5866 (54.9) | |
| Overweight (25–29.9 kg/m$^2$) | 2011 (7.9) | 873 (5.9) | 1138 (10.7) | |
| Obese (≥30 kg/m$^2$) | 577 (2.3) | 196 (1.3) | 381 (3.6) | |
| Middle upper arm circumference (cm) | 23.7±3.2 | 23.3±2.9 | 24.4±3.4 | <0.001 |

*Values are n (%) or mean±SD.
†158 participants reported no longer being married.
‡Asked of ever-married women ($n_{all}$=5834 (3 women did not know); $n_{15–18.9\ years}$=1457; $n_{19–23\ years}$=4377).
§Due to cultural sensitivities, question only asked of ever-married women ($n_{all}$=3451 (28 women did not know); $n_{15–18.9\ years}$=1457; $n_{19–23\ years}$=4377).
¶$n_{all}$=3451 (28 women did not know); $n_{15–18.9\ years}$=567; $n_{19–23\ years}$=2884.
BMI, body mass index; MaPPS, Matiari emPowerment and Preconception Supplementation.

religion, wealth quintile, marital status, reported perception of health and decision-making autonomy (p for model: <0.0001; table 3). While a crude association was observed between BMI categories with food security and self-efficacy, respectively, this was not maintained in the multivariable model (see online supplemental table S2 for all associations and distribution of responses for possible determinants). Compared with those with a normal BMI, those who were underweight were more likely to be from a poorer wealth quintile, non-Muslim, unmarried, and report poorer overall health and participate in less decision-making. Among those overweight/ obese, compared with those with a normal BMI, someone was less likely to be employed as an unskilled manual labourer, but more likely to be from a higher wealth quintile and married.

### Stunting among late adolescent girls and young women

The SDoH risk factors maintained in hierarchical models for stunting differed between late adolescent girls and young women (table 4). Among late adolescent girls, the final model included variables from three levels of the hierarchy (level 1: education, wealth quintile; level 2: food security; level 4: self-efficacy, decision-making; p for model: <0.0001), while for young women, variables from only the two distal-most levels were maintained (level 1: education, wealth quintile; level 2: parity, food security; p for model: <0.0001). The crude association observed

Table 2  Social determinants of health maintained in the hierarchical model of BMI categories (underweight (<18.5 kg/m$^2$), normal weight (18.5–24.9 kg/m$^2$; base) and overweight and obese (≥25 kg/m$^2$)) among late adolescent girls (15–18.9 years) enrolled in the MaPPS Trial at enrolment (n=14 771)*

| Possible determinants | BMI categories | | | | | | | |
|---|---|---|---|---|---|---|---|---|
| | Crude | | | | Multivariable adjusted | | | |
| | Underweight | P value | Overweight/ obese | P value | Underweight | P value | Overweight/ obese | P value |
| Level 1: structural factors—socioeconomic status | | | | | | | | |
| Occupation | | | | | | | | |
| Unskilled manual labour | (Reference) | – | (Reference) | – | (Reference) | – | (Reference) | – |
| Skilled manual labour | 0.85 (0.76 to 0.95) | 0.004 | 2.44 (1.83 to 3.25) | <0.001 | 0.92 (0.79 to 1.07) | 0.26 | 1.83 (1.42 to 2.36) | <0.001 |
| Within the home | 0.83 (0.75 to 0.91) | <0.001 | 3.28 (2.54 to 4.25) | <0.001 | 0.94 (0.83 to 1.07) | 0.34 | 2.03 (1.61 to 2.58) | <0.001 |
| Other† | 0.90 (0.81 to 1.00) | 0.06 | 3.26 (2.48 to 4.30) | <0.001 | 1.14 (0.94 to 1.37) | 0.18 | 1.44 (1.09 to 1.92) | 0.01 |
| Wealth quintile | | | | | | | | |
| Poorest | (Reference) | – | (Reference) | – | (Reference) | – | (Reference) | – |
| Poor | 0.84 (0.76 to 0.94) | 0.002 | 1.47 (1.07 to 2.01) | 0.001 | 0.84 (0.74 to 0.95) | 0.01 | 1.34 (1.07 to 1.67) | 0.01 |
| Middle | 0.79 (0.71 to 0.88) | <0.001 | 2.36 (1.77 to 3.15) | <0.001 | 0.78 (0.68 to 0.89) | <0.001 | 2.07 (1.63 to 2.63) | <0.001 |
| Rich | 0.71 (0.64 to 0.79) | <0.001 | 3.42 (2.59 to 4.51) | <0.001 | 0.69 (0.58 to 0.82) | <0.001 | 2.94 (2.32 to 3.74) | <0.001 |
| Richest | 0.69 (0.62 to 0.77) | <0.001 | 5.18 (3.96 to 6.79) | <0.001 | 0.65 (0.56 to 0.74) | <0.001 | 4.65 (3.49 to 6.20) | <0.001 |
| Level 2: intermediary factors—household and personal characteristics | | | | | | | | |
| Marital status | | | | | | | | |
| Married | (Reference) | | (Reference) | | (Reference) | – | (Reference) | – |
| Unmarried | 1.38 (1.23 to 1.56) | <0.001 | 0.80 (0.66 to 0.97) | <0.03 | 1.42 (1.28 to 1.59) | <0.001 | 0.71 (0.59 to 0.85) | <0.001 |
| Household food security status | | | | | | | | |
| Food insecure | (Reference) | | (Reference) | | (Reference) | – | (Reference) | – |
| Food secure | 0.83 (0.77 to 0.90) | <0.001 | 1.92 (1.61 to 2.28) | <0.001 | 0.91 (0.83 to 0.99) | 0.04 | 1.24 (1.03 to 1.51) | 0.03 |
| Level 3: intermediary factors—health and well-being characteristics | | | | | | | | |
| Perception of own health | | | | | | | | |
| Poor or fair | (Reference) | | (Reference) | | (Reference) | – | (Reference) | – |
| Good | 0.82 (0.74 to 0.92) | 0.001 | 1.23 (0.97 to 1.57) | 0.09 | 0.83 (0.70 to 0.97) | 0.03 | 1.23 (0.89 to 1.68) | 0.21 |
| Excellent | 0.71 (0.63 to 0.80) | <0.001 | 1.40 (1.08 to 1.81) | 0.01 | 0.74 (0.62 to 0.88) | 0.001 | 1.22 (0.93 to 1.60) | 0.16 |
| Level 4: intermediary factors—actions and practices-related characteristics | | | | | | | | |
| Self-efficacy | | | | | | | | |
| Low | (Reference) | – | (Reference) | – | (Reference) | – | (Reference) | – |
| Moderate | 0.85 (0.79 to 0.92) | <0.001 | 1.27 (1.10 to 1.46) | 0.001 | 0.91 (0.85 to 0.97) | <0.001 | 1.09 (0.93 to 1.27) | 0.29 |
| High | 0.75 (0.67 to 0.83) | <0.001 | 1.12 (0.91 to 1.37) | 0.29 | 0.82 (0.72 to 0.92) | <0.001 | 0.91 (0.73 to 1.14) | 0.41 |

*Values are n (%) or RR (95% CI).
†Other category included those who identified themselves as students or professionals.
BMI, body mass index; MaPPS, Matiari emPowerment and Preconception Supplementation; RR, risk ratio.

between stunting and occupation, religion and skipping breakfast, respectively, was not maintained in either multivariable model (see online supplemental table S3). Compared with those non-stunted, the odds were lower that stunted adolescents had any education or came from a wealthier or food secure household; they experienced lower self-efficacy and made fewer autonomous decisions. Similarly, stunted young women were found to have a lower odd of education and being from a food secure household, as well as a higher odd of participating in unskilled manual labour; once the other variables were considered in the model, they were more likely not to have been pregnant compared with those non-stunted.

## DISCUSSION

Using enrolment data from a large trial conducted in rural Pakistan, we found high levels of undernutrition among late adolescent and young women 15–23 years using anthropometric indicators. Our study offers the most comprehensive analysis of the determinants of malnutrition among late adolescent girls and young

**Table 3** Social determinants of health maintained in the hierarchical model of BMI categories (underweight (<18.5 kg/m²), normal weight (18.5–24.9 kg/m²; base) and overweight and obese (≥25 kg/m²)) among young women (19–23 years) enrolled in the MaPPS Trial at enrolment (n=10 676)*

| | BMI categories | | | | | | | |
| --- | --- | --- | --- | --- | --- | --- | --- | --- |
| | Crude | | | | Multivariable adjusted | | | |
| Possible determinants | Underweight | P value | Overweight/obese | P value | Underweight | P value | Overweight/obese | P value |
| Level 1: structural factors—socioeconomic status | | | | | | | | |
| Occupation | | | | | | | | |
| Unskilled manual labour | (Reference) | – | (Reference) | – | (Reference) | – | (Reference) | – |
| Skilled manual labour | 0.81 (0.71 to 0.93) | 0.002 | 2.77 (2.17 to 3.55) | <0.001 | 0.95 (0.83 to 1.09) | 0.48 | 1.91 (1.48 to 2.48) | <0.001 |
| Within the home | 0.75 (0.67 to 0.84) | <0.001 | 3.52 (2.81 to 4.42) | <0.001 | 0.93 (0.83 to 1.05) | 0.25 | 2.07 (1.61 to 2.67) | <0.001 |
| Other† | 0.76 (0.62 to 0.93) | 0.007 | 3.80 (2.83 to 5.11) | <0.001 | 1.11 (0.83 to 1.49) | 0.47 | 1.64 (1.19 to 2.26) | 0.002 |
| Religion | | | | | | | | |
| Non-Muslim | (Reference) | – | (Reference) | – | (Reference) | – | (Reference) | – |
| Muslim | 0.64 (0.56 to 0.74) | <0.001 | 1.74 (1.35 to 2.25) | <0.001 | 0.76 (0.61 to 0.93) | 0.01 | 0.98 (0.71 to 1.36) | 0.92 |
| Wealth quintile | | | | | | | | |
| Poorest | (Reference) | – | (Reference) | – | (Reference) | – | (Reference) | – |
| Poor | 0.81 (0.71 to 0.92) | 0.002 | 1.98 (1.48 to 2.64) | <0.001 | 0.87 (0.74 to 1.02) | 0.09 | 1.81 (1.36 to 2.43) | <0.001 |
| Middle | 0.73 (0.64 to 0.83) | <0.001 | 3.31 (2.52 to 4.34) | <0.001 | 0.79 (0.70 to 0.89) | <0.001 | 2.84 (2.02 to 4.00) | <0.001 |
| Rich | 0.62 (0.54 to 0.71) | <0.001 | 4.68 (3.60 to 6.09) | <0.001 | 0.68 (0.59 to 0.78) | <0.001 | 3.88 (2.85 to 5.28) | <0.001 |
| Richest | 0.55 (0.48 to 0.63) | <0.001 | 6.36 (4.92 to 8.22) | <0.001 | 0.59 (0.48 to 0.72) | <0.001 | 5.24 (3.93 to 6.98) | <0.001 |
| Level 2: intermediary factors—household and personal characteristics | | | | | | | | |
| Marital status | | | | | | | | |
| Married | (Reference) | – | (Reference) | – | (Reference) | – | (Reference) | – |
| Unmarried | 1.14 (1.04 to 1.24) | 0.004 | 0.90 (0.81 to 1.01) | 0.08 | 1.25 (1.11 to 1.42) | <0.001 | 0.77 (0.69 to 0.85) | <0.001 |
| Perception of own health | | | | | | | | |
| Poor or fair | (Reference) | – | (Reference) | – | (Reference) | – | (Reference) | – |
| Good | 0.78 (0.68 to 0.89) | <0.001 | 1.18 (0.97 to 1.44) | 0.09 | 0.78 (0.69 to 0.88) | <0.001 | 1.16 (0.94 to 1.44) | 0.18 |
| Excellent | 0.75 (0.64 to 0.87) | <0.001 | 1.35 (1.09 to 1.67) | 0.006 | 0.78 (0.69 to 0.89) | <0.001 | 1.16 (0.89 to 1.54) | 0.27 |
| Decision-making autonomy | | | | | | | | |
| All decisions made by family | (Reference) | – | (Reference) | – | (Reference) | – | (Reference) | – |
| Most decisions made by family | 0.91 (0.83 to 1.01) | 0.08 | 1.08 (0.95 to 1.24) | 0.23 | 0.92 (0.85 to 0.99) | 0.03 | 1.08 (0.93 to 1.27) | 0.31 |
| Decisions made jointly with family or autonomously | 0.85 (0.76 to 0.94) | 0.003 | 1.19 (1.04 to 1.37) | 0.01 | 0.85 (0.77 to 0.95) | 0.003 | 1.24 (1.06 to 1.45) | 0.01 |

*Values are n (%) or RR (95% CI).
†Other category included those who identified themselves as students or professionals.
BMI, body mass index; MaPPS, Matiari emPowerment and Preconception Supplementation; RR, risk ratio.

women in rural Pakistan. Across the multivariable models for acute and chronic measures of undernutrition, those who were impoverished were at increased risk regardless of age group. This suggests that structural factors play an important, underlying role in female nutrition in this setting. That an emerging burden of overweight/obesity was also observed in this rural population indicates that interventions are needed to address both ends of the malnutrition spectrum. To the best of our knowledge, this is the largest survey of a cohort that includes both

unmarried and married Pakistani late adolescent girls and young women.

Pakistan is known to have a high burden of malnutrition, with a long term impact on human capital and costing an estimated US$7.6 billion/year.[34] We found the mean BMI among study participants was lower than the PDHS national average (20.2 kg/m² vs 25.7 kg/m²).[16] This likely reflects that the PDHS only surveyed evermarried WRA (15–49 years). In the recently completed 2018 Pakistan National Nutrition Survey, 22.6%, 19.8%

**Table 4** Comparison of social determinants of health in hierarchical model of stunting (height <145 cm) among late adolescent girls (n=14 771) and adult women (n=10 676) enrolled in the MaPPS Trial at enrolment

| Possible determinants | Stunting among late adolescent girls | | | | Stunting among young women | | | |
| --- | --- | --- | --- | --- | --- | --- | --- | --- |
| | Crude | | Multivariable adjusted | | Crude | | Multivariable adjusted | |
| | OR (95% CI) | P value | OR (95% CI) | P value | OR (95% CI) | P value | OR (95% CI) | P value |
| Level 1: structural factors—socioeconomic status | | | | | | | | |
| Highest level of education | | | | | | | | |
| None | (Reference) | – | (Reference) | – | (Reference) | – | (Reference) | – |
| Primary | 0.65 (0.57 to 0.74) | <0.001 | 0.76 (0.62 to 0.94) | 0.01 | 0.62 (0.51 to 0.74) | <0.001 | 0.76 (0.63 to 0.93) | 0.006 |
| Secondary or higher | 0.33 (0.29 to 0.38) | <0.001 | 0.48 (0.39 to 0.59) | <0.001 | 0.51 (0.43 to 0.61) | <0.001 | 0.79 (0.66 to 0.95) | 0.01 |
| Wealth quintile | | | | | | | | |
| Poorest | (Reference) | – | (Reference) | – | (Reference) | – | (Reference) | – |
| Poor | 0.83 (0.72 to 0.97) | 0.02 | 0.91 (0.77 to 1.08) | 0.29 | 0.88 (0.72 to 1.08) | 0.22 | 0.92 (0.73 to 1.17) | 0.52 |
| Middle | 0.67 (0.57 to 0.78) | <0.001 | 0.79 (0.65 to 0.96) | 0.02 | 0.65 (0.52 to 0.80) | <0.001 | 0.70 (0.59 to 0.84) | <0.001 |
| Rich | 0.46 (0.39 to 0.55) | <0.001 | 0.63 (0.49 to 0.81) | <0.001 | 0.48 (0.38 to 0.60) | <0.001 | 0.54 (0.40 to 0.74) | <0.001 |
| Richest | 0.26 (0.21 to 0.32) | <0.001 | 0.42 (0.35 to 0.51) | <0.001 | 0.34 (0.27 to 0.43) | <0.001 | 0.41 (0.29 to 0.57) | <0.001 |
| Level 2: intermediary factors—household and personal characteristics | | | | | | | | |
| Ever been pregnant | | | | | | | | |
| Yes | (Reference) | – | – | – | (Reference) | – | (Reference) | – |
| No | 0.97 (0.74 to 1.27) | 0.81 | – | – | 1.13 (0.96 to 1.32) | 0.15 | 1.27 (1.10 to 1.47) | 0.001 |
| Household food security status | | | | | | | | |
| Food insecure | (Reference) | – | (Reference) | – | (Reference) | – | (Reference) | – |
| Food secure | 0.58 (0.51 to 0.64) | <0.001 | 0.77 (0.66 to 0.90) | 0.001 | 0.57 (0.49 to 0.66) | <0.001 | 0.73 (0.59 to 0.90) | 0.003 |
| Level 4: intermediary factors—actions and practices-related characteristics | | | | | | | | |
| Self-efficacy | | | | | | | | |
| Low | (Reference) | – | (Reference) | | (Reference) | | – | – |
| Moderate | 0.62 (0.56 to 0.70) | <0.001 | 0.74 (0.64 to 0.84) | <0.001 | 0.83 (0.71 to 0.98) | 0.03 | – | – |
| High | 0.59 (0.50 to 0.71) | <0.001 | 0.74 (0.62 to 0.87) | <0.001 | 0.78 (0.63 to 0.97) | 0.03 | – | – |
| Decision-making autonomy | | | | | | | | |
| All decisions made by family | (Reference) | – | (Reference) | – | (Reference) | – | – | – |
| Most decisions made by family | 0.83 (0.74 to 0.94) | 0.003 | 0.88 (0.76 to 1.01) | 0.08 | 0.99 (0.84 to 1.16) | 0.90 | – | – |
| Decisions made jointly with family or autonomously | 0.74 (0.62 to 0.89) | 0.001 | 0.76 (0.62 to 0.92) | 0.01 | 0.85 (0.71 to 1.02) | 0.09 | – | – |

MaPPS, Matiari emPowerment and Preconception Supplementation.

and 10.2% of WRA in Sindh province were found to be underweight, overweight and obese, respectively.[35] Notably, both the PDHS and National Nutrition Survey include urban settings, where the risk of overweight and obesity is higher.[36]

Of the limited adolescent nutrition studies available globally, wealth has consistently been found to be a determinant of undernutrition in multiple settings.[37–41] Regardless of whether someone was an adolescent or young women, we found that at least one structural SDoH variable tied to poverty was associated with an increased risk of undernutrition outcomes. Notably, food insecurity

was maintained in several models. At least one variable considered a proxy for empowerment was maintained in each adolescent model, with greater empowerment being associated with an improvement in nutritional status. Considered a blend of resources (materials that allow someone to make her goals a reality), agency (the ability to define and act on someone's goals) and achievement (obtaining a desired goal), empowerment enables someone to make decisions to take charge of her own life and affect her personal circumstances.[42] Empowered women are widely recognised to be better able to promote their own health and well-being; whereas,

lacking empowerment is suggested to present a further barrier to achieving optimal nutritional status.[3]

We had hypothesised that married late adolescent and young women might be at greater risk for undernutrition because early marriage has been suggested to increase nutritional vulnerability[43]; however, in the hierarchical models for both adolescent and young women, being married was associated with increased likelihood of being in a higher BMI category compared with those unmarried. This it is likely attributable to an increase in status within the household, as married participants contributed more to food-related decision-making (data not shown). An association between marital status and stunting, a chronic measure of poor nutrition across the lifespan, was not observed.

There is a recognised need in Pakistan to focus on the health and nutrition of adolescent girls,[44 45] and it is well documented that preconception health and well-being are important to the individual, as well as to her future offspring.[46] To address inequalities due to poverty, the use of social safety nets could function as an important resource. As food insecurity prevalence was notable, provision of cash transfers, for example, could lend to the purchase of more nutritious food, although we are not certain at this time of the exact effect that this might have on an adolescent or young women's actual intake. However, economic status alone does not confer a healthy BMI.[41] We would note that there will need to be a concurrent emphasis on maintaining a healthy BMI, given that the prevalence of overweight/obesity in Pakistan has increased over time in both urban and rural settings, the existing double burden of malnutrition, and ongoing nutrition transition. To address structural and empowerment-related barriers, increased and coordinated efforts between the public health system and the education sector could have potential. The Government of Pakistan has recently released its nutritional strategy and operational plan for targeting adolescent malnutrition, which will be important to implementing legislation on adolescent nutrition.[47]

Limitations of this study include that our catchment area was restricted to the areas of Matiari District covered by the LHW Programme. Furthermore, our findings are specific to the context in rural Pakistan, and findings may not reflect settings with different cultural norms or burdens of undernutrition, as seen in Africa and South America.[48] While BMI is a globally accepted index of nutritional status, the relationship between BMI and body composition can be controversial, as BMI does not account for excess weight due to adiposity versus muscularity, oedema, or bone mass.[6] It would have been informative to have had additional measures of body composition given its influence on health. The SDoH considered within this study were proxy variables and required self-reporting, thus there could be measurement error (eg, social desirability bias). It is also possible that all SDoH relevant to context were not considered, although the assessed explanatory variables were informed by the literature.

We additionally acknowledge that the adolescent period does commence at 10 years, although given the aims of the MaPPS Trial we did not enrol adolescent girls 10–14 years. The larger study focused on adolescent and young women, not adolescent boys or young men; yet, males are also important given their earning potential and contribution to the economy.[45] A further similar evaluation of this group would be informative, and aid understanding gendered cultural norms. Although the assessment of macrolevel SDoH (eg, governance, policies) was beyond the scope of this assessment, the importance of structural SDoH to all models suggests that underlying factors that create divisions within society are precarious. In their cross-sectional assessment of the factors affecting thinness among adolescent girls 10–19 years in low-income and middle-income countries, Candler et al[37] found that macroeconomic factors contribute to the genesis of adolescent thinness, and that reducing adolescent thinness should include addressing national food security and economic development.

## CONCLUSIONS

Given the findings of this study, and the concerning status of malnutrition in Pakistan, commitment to improving the nutrition of adolescent and young women in Pakistan is critical and should include those unmarried. Reaching these individuals is important not only because of their reproductive potential, but also given their right to optimal nutrition and well-being.[49] To target undernutrition, particularly, there is a need for both direct and indirect nutrition and empowerment interventions that are integrated and multisectoral,[50] as highlighted in the recent national adolescent nutrition policy recommendations.[47] An ideal programme for adolescent and young women should be inclusive and accommodating, reach those who are most vulnerable, and account for cultural-relevant and gender-relevant factors.[51]

**Acknowledgements** The authors wish to acknowledge the participants in the MaPPS Trial and the field research team. We are grateful for the ongoing data-related assistance and support provided by Rasool Bux, Arjumand Rizvi and the members of the Data Management Unit at Aga Khan University.

**Contributors** J-ABB and ZAB designed research; YW and SBS conducted research; IA and AH performed data management; J-ABB analysed data; J-ABB wrote first manuscript draft; ZAB, YW, SBS, IA and AH provided manuscript feedback; ZAB has primary responsibility for final content as the guarantor. All authors reviewed this manuscript and approved the final version.

**Funding** The MaPPS Trial was supported by the Bill and Melinda Gates Foundation (grant number: OPP1148892) and the World Food Programme (grant number: HQ15NF493 - CTR). J-ABB received research scholarships from the Vanier Canada Graduate Scholarship; Hospital for Sick Children Foundation Student Scholarship Program and the Province of Ontario.

**Competing interests** None declared.

**Patient and public involvement** Patients and/or the public were involved in the design, or conduct, or reporting, or dissemination plans of this research. Refer to the Methods section for further details.

**Patient consent for publication** Not applicable.

**Ethics approval** Ethical review for this study was obtained from the Aga Khan University Ethics Review Committee (Protocol #4324-Ped-ERC-16) and the Hospital for Sick Children Research Ethics Board (Protocol #1000054682).

**Provenance and peer review** Not commissioned; externally peer reviewed.

**Data availability statement** Data are available on reasonable request. The anonymised, individual-level data from this cross-sectional assessment, situated within a clinical trial, will be made available on reasonable request from the corresponding author.

**ORCID iDs**
Jo-Anna B Baxter http://orcid.org/0000-0002-4718-0382
Sajid B Soofi http://orcid.org/0000-0003-4192-8406

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
