## [Reviewer comments · BMJ Open]

ARTICLE DETAILS

TITLE (PROVISIONAL)	Drivers of malnutrition among late adolescent and young women in rural Pakistan: A cross-sectional assessment of the MaPPS Trial
AUTHORS	Baxter, Jo-Anna; Wasan, Yaqub; Hussain, Amjad; Soofi, Sajid; Ahmed, Imran; Bhutta, Zulfiqar

VERSION 1 – REVIEW

REVIEWER	Fledderjohan, Jasmine Lancaster University, Sociology
REVIEW RETURNED	08-Dec-2022

GENERAL COMMENTS	In this paper, the authors use MaPPS data to understand the social determinants of health associated with malnutrition for young women in Pakistan. It is written clearly, and this is an important topic. I have several queries for the authors: 1. I understand that the data have been described in detail elsewhere, but a few more details would be helpful in the methods section here. In particular: o It's a little unclear as written whether the data the team are analysing are primary or secondary data. Please clarify.o Were all households in each cluster that met the screening criteria selected for eligible women to be invited to participate, or was there a process of selection of households within each cluster?o Given literacy rates in the adult population in Pakistan (and given the high proportion reporting no education in Table 1), it seems very likely at least some participants may have limited literacy. How was written consent taken in this context?o There are many possible scales that could have been deployed to measure some of the variables, such as food insecurity, depression, and self-efficacy. More information is needed to understand how these variables were measured in this study please. 2. I also have some queries regarding the modelling decisions o It's unclear why the authors generated separate models for each age group rather than using age group as a control in the regression models?o There seems to be strong potential for reciprocal relationships for some of the variables (most prominent among these is self-rated health, but also for things like depression and eating behaviours). (How) have the authors accounted for this?o The use of stepwise modelling is concerning. There is a large literature (dating back at least to the 1970's, inc. Lewis-Beck's
---

	work on this, Gary King’s work in the 1980’s, etc.) highlighting the serious limitations of stepwise modelling. It is also unclear why the authors took this approach since they referenced a previous conceptual model and suggest they had a clear theoretical reason for including the variables they had selected—why omit variables you have a strong theoretical reason to believe matter in this context? This also makes it impossible to compare the coefficients for the variables included in the models for adolescent girls vs. young women because the models control for different factors.
--	--

VERSION 1 – AUTHOR RESPONSE

Responses to Reviewer 1

Manuscript ID: bmjopen-2022-063734

Date of submission: January 15, 2023

Reviewer Comment	Response
1) Questions about Methods	
It’s a little unclear as written whether the data the team are analysing are primary or secondary data. Please clarify.	Thank you for the requested clarification. The analyses described in the manuscript have been conducted using enrolment data only from the MaPPS Trial, which was collected by our team collected. At the start of the ‘Methods’ section, we had included the text “This study used cross-sectional data from late adolescent girls and young women (15–23 y) collected at enrolment within the Matiari emPowerment and Preconception Supplementation (MaPPS) Trial.” All members of our team were involved in different aspects of the design, conduction, and management of the data from the MaPPS Trial. To improve the clarity of primary trial outcome, versus the outcome measure for this study, we have added “(primary trial outcome)” in the section ‘Setting and study design’ after the mention of low-birth-weight births. Additionally, the sentence first under ‘Outcomes of interest’ has been revised to read “The outcomes of interest in the present study were cut-offs for BMI...”
Were all households in each cluster that met the screening criteria selected for eligible women to be invited to participate, or was there a process of selection of households within each cluster?	All households in each cluster covered by the LHW Programme were eligible to be included the household listing exercise. This household listing step is consistent with AKU’s existing surveillance program within Matiari district. The screening criteria were applied after this. To further clarify, we have revised the text as follows, “Late adolescent girls and young women 15–23 y were identified from a household listing exercise conducted from December 2016 to May 2017, wherein all households”

	covered by LHWs in a cluster unit were surveyed by study personnel. To screen for participation in the MaPPS Trial, late adolescent girls and young women 15–23 y were approached at their homes by study personnel to further assess their eligibility and ≥1 participant could be enrolled per household.
Given literacy rates in the adult population in Pakistan (and given the high proportion reporting no education in Table 1), it seems very likely at least some participants may have limited literacy. How was written consent taken in this context?	We thank the reviewer for this question. As they have noted, literacy rates were low in the study area. As such, the informed consent process included providing participants with a written consent form and a verbal explanation of the purpose and voluntary nature of the MaPPS Trial; participation components; and potential benefits and harms by study personnel. Written documentation of informed consent was obtained either as a signature or a left thumb impression in the case that a participant was illiterate. A witness was required in the case that thumb impression was required. To clarify this, we have revised the text as follows “The purpose and voluntary nature of the trial was verbally explained following confirmation of eligibility by study personnel and a written consent form was provided. Written documentation of informed consent was obtained as either a signature or a thumb impression, in the case that a participant was illiterate.”
There are many possible scales that could have been deployed to measure some of the variables, such as food insecurity, depression, and self-efficacy. More information is needed to understand how these variables were measured in this study please.	We thank you for their attention to variable measurement. Within the ‘Explanatory variables’ section, after each variable is mentioned in the manuscript text, a numeric reference had been provided. These refer the reader to the corresponding tool used for assessment. We used the tools as described by the authors in the cited articles to the best extent possible. In some instances, because few responses were observed (e.g., extremely severe depression, severe depression), we grouped these responses (e.g., extreme or severe depression), as shown in the Supplementary Material. In instances where we categorized a variable differently (i.e., self-efficacy score, decision-making autonomy), this was explained in-text. To further clarify the exact tools we used in-text, we have added the following details, “Other health-related proxies included self-reported health status and mental health (experience of depression, anxiety, and stress-related emotions over the preceding week; Depression, Anxiety, and Stress Score – 21 Items).²⁸ Other nutrition-related proxies included household food insecurity (Household Food Insecurity Access Scale)²⁹ and skipping breakfast and sharing dinner with family (Health Behaviour in School-aged Children

	survey).³⁰ Empowerment proxies included generalized self-efficacy (Generalized Self-Efficacy Scale)³¹ and decision-making autonomy (PDHS).¹⁷
2) Modelling decisions	
It's unclear why the authors generated separate models for each age group rather than using age group as a control in the regression models?	We found that there was a difference in the prevalence of both BMI categories and stunting between late adolescent girls and young women, so we generated a separate model for each age group. This was noted in-text within the second paragraph of the 'Statistical analyses' section. Our decision was also grounded in that The WHO and UNICEF report on the anthropometrics of adolescent girls up to 19 y, and we felt that separate presentation of the models was important to the interpretability and comparability with other resources for policy makers. This includes WHO (2017)'s Global Accelerated Action for the Health of Adolescents (AA-HA!) and WHO (2018)'s Guideline: Implementing effective actions for improving adolescent nutrition, which are both important guidelines for countries for implementing nutrition interventions targeted to adolescents, especially in countries with a large prevalence of undernutrition. Within Pakistan, specifically, these guidelines have informed the existing Pakistan Adolescent Nutrition Strategy and Operational Plan (Government of Pakistan 2020). To further clarify our rationale, we have added the following text within the 'Statistical analyses' section, "Because we found that there was a difference in the prevalence of both BMI categories and stunting between late adolescent girls and young women, and to be consistent with standard age-based reporting of anthropometric measures, a separate model was generated for each age group." References: Government of Pakistan. Pakistan Adolescent Nutrition Strategy and Operational Plan. Islamabad: Government of Pakistan; 2020. WHO. Global Accelerated Action for the Health of Adolescents (AA-HA!): guidance to support country implementation. Geneva: WHO; 2017. WHO. Guideline: implementing effective actions for improving adolescent nutrition. Geneva: WHO; 2018.
There seems to be strong potential for reciprocal relationships for some of the variables (most prominent among these is self-rated health, but also for things like depression and eating behaviours). (How) have the authors accounted for this?	We thank the reviewer for their attention to this. We have described our hierarchical modelling approach a bit more below, but briefly we did assess multicollinearity of the variables considered for the model. This was done using the variance inflation factor (vif) command in Stata and employed a cut-off value of ≥ 5.

	To clarify that we did investigate this, the following text has been added, “Multicollinearity between variables was assessed using the variance inflation factor (VIF), although we did not find that multicollinearity affected the selected variables using the criteria $VIF \geq 5$.”
The use of stepwise modelling is concerning. There is a large literature (dating back at least to the 1970’s, inc. Lewis-Beck’s work on this, Gary King’s work in the 1980’s, etc.) highlighting the serious limitations of stepwise modelling. It is also unclear why the authors took this approach since they referenced a previous conceptual model and suggest they had a clear theoretical reason for including the variables they had selected—why omit variables you have a strong theoretical reason to believe matter in this context? This also makes it impossible to compare the coefficients for the variables included in the models for adolescent girls vs. young women because the models control for different factors.	We thank the reviewer for their concern about the model building approach and agree that the use of a stepwise modelling approach is not robust. We hope that the following explanation clarifies the method employed, as the primary approach was a hierarchical modelling, as cited in the manuscript (Victora et al 1997 [34]). This method is intended for multivariate data analysis in epidemiological studies where determinants of a condition are sought. Using this approach allowed us to address the complex hierarchical inter-relationships between variables, and was generated with the expert guidance of a statistician. This differs from a standard stepwise regression, in that precedence is given to the distal-most variables to be able to understand determinant exposures, as opposed to treating all exposures as related to the outcome (as in stepwise modelling). As noted by the reviewer, we developed a conceptual model, which has been previously published and was cited in the manuscript (Baxter et al 2021 [25]). We used the conceptual model as a guide for our hierarchical model, and hypothesized that there were four levels of determinants underlying malnutrition (socioeconomic status; household and personal factors; health and well-being; and actions and practices. To improve the flow of the text in the section ‘Statistical Analyses’ we have moved these details to paragraph 3 and provided some additional clarifying text. The text now reads, “The multi-level model building approach described by Victora et al³⁴ was followed to generate an adapted hierarchical model with 4-levels. Guided by our conceptual model,²⁵ we hypothesized that there were four levels of SDoH explanatory variables within the hierarchy: (1) socioeconomic status (distal-most level): education, occupation, religion, and wealth quintile; (2) household and personal factors: marital status, parity, and household food security; (3) health and well-being: perception of own health, experience of depression-, anxiety-, and stress-like feelings; and (4) actions and practices (proximal-most level): self-efficacy, participation in decision making, skipping breakfast, and eating dinner with family. For all analyses, the reference for each SDoH explanatory variable was set as the category hypothesized to present the greatest nutritional vulnerability.^{25”}

Within the hierarchical approach, the variables in the distal-most level of the model are entered first (i.e., socioeconomic status). Given we hypothesized four levels, this included four steps. To find a model that best explained the variability in the data, and so that we did not over fit the model, we did then employ a backward-stepwise model building approach at each level of the hierarchy. This allowed us to investigate the relationships between variables at each level, as we observed the effect of serial removal of variables on β -coefficients and standard errors. Any variables already identified as relevant to model in a previous level of the hierarchy were maintained, as is consistent with the hierarchical modelling approach.

We did aim to be transparent about the distribution of responses between the two age groups and our modelling exercise, as per our inclusion of Supplemental Material. This has been referenced throughout the Results section. Within the Supplemental Material, we presented the distribution of responses for all variables (as opposed to just those maintained in the final models, which are shown in the main manuscript) and all univariate analyses for all variables (shown under the heading 'Crude'). Ultimately, we did not find that the variables that best fit the model were identical between adolescent girls and young women, and consistent with the hierarchical modelling approach maintained those the best explained the variability in the data.

We hope that this explanation and implemented changes have helped to clarify the modelling approach taken.

Reference:

Victora CG, Huttly SR, Fuchs SC, et al. The role of conceptual frameworks in epidemiological analysis: a hierarchical approach. *Int J Epidemiol* 1997;26:224-27.
Baxter JB, Wasan Y, Hussain A, Soofi SB, Ahmed I, Bhutta ZA. Characterizing Micronutrient Status and Risk Factors among Late Adolescent and Young Women in Rural Pakistan: A Cross-Sectional Assessment of the MaPPS Trial. *Nutrients*. 2021;13(4):1237.